# High-harmonic generation by field enhanced femtosecond pulses in metal-sapphire nanostructure

Seunghwoi Han[1,*], Hyunwoong Kim[1,*], Yong Woo Kim[1], Young-Jin Kim[2], Seungchul Kim[3], In-Yong Park[4] & Seung-Woo Kim[1]

Plasmonic high-harmonic generation (HHG) drew attention as a means of producing coherent extreme ultraviolet (EUV) radiation by taking advantage of field enhancement occurring in metallic nanostructures. Here a metal-sapphire nanostructure is devised to provide a solid tip as the HHG emitter, replacing commonly used gaseous atoms. The fabricated solid tip is made of monocrystalline sapphire surrounded by a gold thin-film layer, and intended to produce EUV harmonics by the inter- and intra-band oscillations of electrons driven by the incident laser. The metal-sapphire nanostructure enhances the incident laser field by means of surface plasmon polaritons, triggering HHG directly from moderate femtosecond pulses of $\sim 0.1\,\mathrm{TW\,cm}^{-2}$ intensities. The measured EUV spectra exhibit odd-order harmonics up to $\sim 60\,\mathrm{nm}$ wavelengths without the plasma atomic lines typically seen when using gaseous atoms as the HHG emitter. This experimental outcome confirms that the plasmonic HHG approach is a promising way to realize coherent EUV sources for nano-scale near-field applications in spectroscopy, microscopy, lithography and atto-second physics.

[1] Department of Mechanical Engineering, Korea Advanced Institute of Science and Technology (KAIST), 291 Daehak-ro, Yuseong-gu, Daejeon 305-701, South Korea. [2] School of Mechanical and Aerospace Engineering, Nanyang Technological University (NTU), 50 Nanyang Avenue, Singapore 639798, Singapore. [3] Max Planck Center for Attosecond Science, Max Planck POSTECH/KOREA Res. Initiative, Pohang, Gyeongbuk 376-73, South Korea. [4] Division of Industrial Metrology, Korea Research Institute of Standards and Science (KRISS), Daejeon 305-340, South Korea. * These authors equally contributed to this work. Correspondence and requests for materials should be addressed to S.-W.K. (email: swk@kaist.ac.kr).

High-harmonic generation (HHG) is a coherent frequency-conversion process arising from gaseous atoms[1–4] or crystalline solids[5–7] when they are irradiated by an intense laser pulse. This nonlinear process permits generating extreme ultraviolet and X-ray radiation from femtosecond laser pulses of infrared or visible wavelengths. The laser pulses employed in HHG need to be able to deliver strong intensities of $\sim 10\,TW\,cm^{-2}$, thus chirped pulse amplification (CPA) is commonly adopted to raise the peak power of femtosecond pulses emanating from an oscillator[8,9]. Recently efforts have been made to take advantage of the strong field enhancement that occurs in nanostructures to attain the field intensities needed for HHG directly from a moderate-power laser oscillator. This plasmonic nanostructure-assisted HHG is intended to build a compact extreme ultraviolet or X-ray source by removing the bulky chirped pulse amplification process, and more importantly, seek to establish a nano-scale basis for conducting spectroscopy[10], microscopy[11], lithography[12] and atto-second physics[13], ultimately on a single-molecule level.

Theoretical studies have suggested that strong field enhancement for HHG may be achievable with various nanostructure designs such as nano-antennas[14–16] and tapered waveguides[17]. Experimental investigations of plasmonic HHG have been carried out by devising bow-tie and funnel-waveguide nanostructures[18–23]. The nanostructures adopted thus far have suffered from two major shortcomings. One is the low resistance to thermal damage caused by the incident laser pulses, which leads to structural deformation and/or vapourization, shortening the nanostructure lifetime span[24]. The other is that the hot-spot volume that provides the strong field enhancement for HHG is narrowly localized. This apparently limits the harmonic yield when using gaseous atoms as the HHG emitter and consequently the plasma atomic lines overpower the harmonics[25–27]. Increasing the gas supply pressure results in no noticeable effect in multiplying the emitter density in the hot-spot volume because gaseous atoms rapidly disperse into vacuum by adiabatic expansion[28].

In this investigation, we designed a nanostructure that replaces gaseous atoms with a monocrystalline sapphire tip that functions as a solid HHG emitter. The nanostructure is a three-dimension conic waveguide that yields strong field enhancement on the sapphire tip along the metal-sapphire interface by means of surface plasmon polaritons. The monocrystalline structure of the sapphire tip contributes to HHG by means of the intra- and/or inter-band oscillations of electrons driven by the enhanced laser field. Finite-difference time-domain (FDTD) simulation predicts that the intensity enhancement occurring on the sapphire tip reaches $\sim 20\,dB$ without significant degradation of the ultrafast temporal characteristics of the original pulse. Experimental data clearly shows high-harmonic peaks up to the thirteenth-order using a laser oscillator emitting moderate pulses of $\sim 0.1\,TW\,cm^{-2}$ intensities. This result validates that the metal-sapphire nanostructure devised in this study successfully copes with the difficulty of securing sufficient gaseous atoms within the

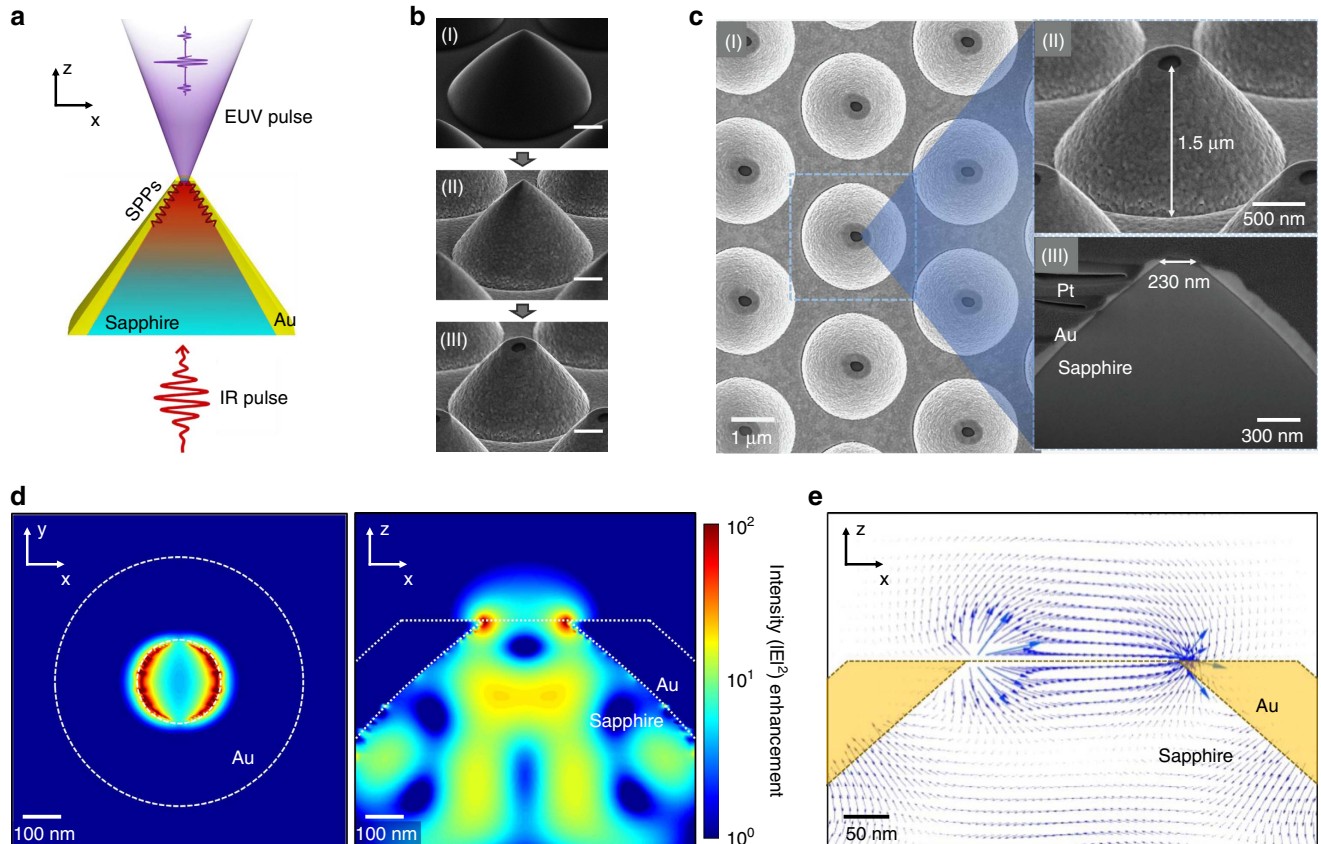

**Figure 1 | Fabrication of the metal-sapphire nanostructure. (a)** Concept of field enhancement by means of surface plasmon polaritons (SPPs) created along the metal-sapphire interface. (**b**) Scanning electron microscopy (SEM) images showing fabrication processes in sequence (scale bar, 500 nm); (I) sapphire patterning by dry plasma etching, (II) gold layer deposition by chemical vapour deposition and (III) exit aperture milling by focused ion beam. (**c**) Magnified SEM images with dimensions. The cutout image (III) was made with the focused ion beam milling process taken after Pt deposition over the entire nanostructure. (**d**) Finite-difference time-domain (FDTD) calculation result for the enhanced intensity distribution over the xy-plane on the sapphire tip (left) and the xz-plane of the cross-section (right). (**e**) Calculated electric field near the edges of the sapphire tip on the xz-plane.

small hot-spot volume by employing the monocrystalline solid as the HHG emitter.

## Results

**Nanostructure design and fabrication.** Figure 1 shows the metal-sapphire nanostructure that was prepared in this study to generate extreme ultraviolet harmonics by plasmonic field enhancement. The incident femtosecond pulse propagates along the metal-sapphire interface up to the cone-shaped sapphire tip (Fig. 1a). As illustrated in the scanning electron microscopy images (Fig. 1b), the nanostructure is fabricated in three distinct steps: First, a sapphire cone is formed by plasma dry etching on a 430 μm thick monocrystalline sapphire substrate (0001). The resulting conic shape has a height of 1.5 μm from the bottom to the apex, and a 2.4 μm base diameter at the bottom. Second, a 3 nm Cr adhesion layer is applied, and an Au thin-film layer is

deposited over the entire surface of the sapphire cone by chemical vacuum deposition to a 200 nm thickness on top of the adhesion layer. Third, the Au-covered apex is flattened by focused ion beam milling so that the sapphire tip inside is exposed with an exit aperture of 230 nm diameter (Fig. 1c).

**FDTD analysis.** In accordance with the actual experimental set-up (Fig. 2a), the FDTD calculation assumed that the nanostructure was illuminated from the bottom base side by an incident laser pulse having an 800 nm center wavelength with 12 fs duration (see Methods). The polarization direction of the incident laser pulse is aligned parallel to the x axis of the circular cross-section (Fig. 1d). The calculation result shows that the enhanced intensity field appears to be symmetrical about the laser polarization direction. The maximum field enhancement occurs along the sapphire tip on the exit aperture where the

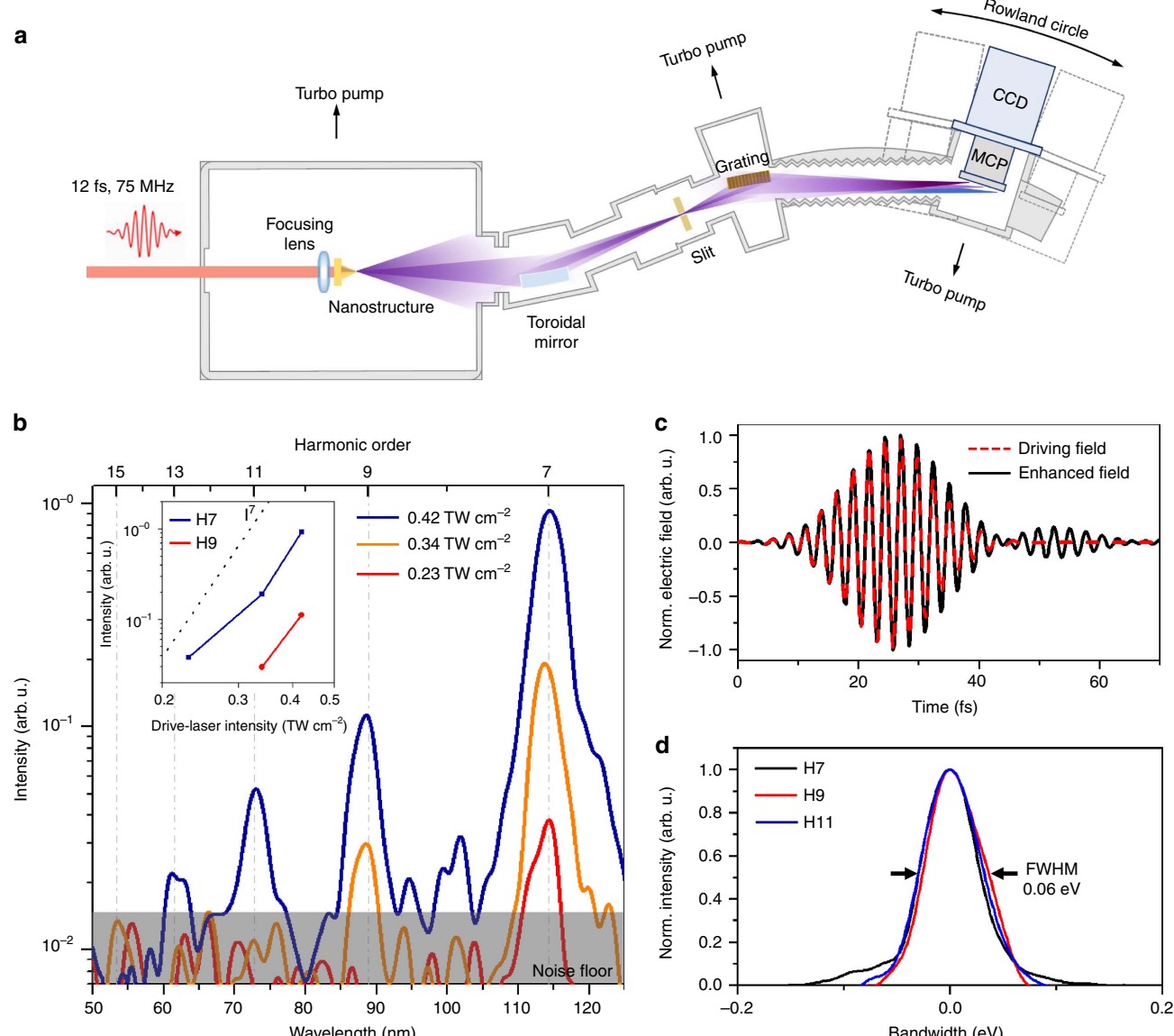

**Figure 2 | High-harmonic generation from the metal-sapphire nanostructure.** (**a**) Overall hardware configuration for extreme ultraviolet generation and spectrum measurement. (**b**) Measured extreme ultraviolet spectra showing HHG peaks. Each spectrum was acquired during 30 s and its raw data was low-pass filtered in the wavelength domain with a 1.0 nm cutoff for smoothening. Harmonic yield vs driving laser intensity (inset). The peak seen between H7 and H9 is reckoned as an even-order harmonic of H8 generated by the steep variation of the enhanced electric field[16] or due to unknown noise. (**c**) FDTD-simulated temporal profile of the enhanced field at the sapphire tip for the incident laser field. (**d**) Normalized profiles of measured HHG peaks. The bandwidth represents the photon energy spread of each peak divided by its harmonic order. FWHM: full-width at half-maximum.

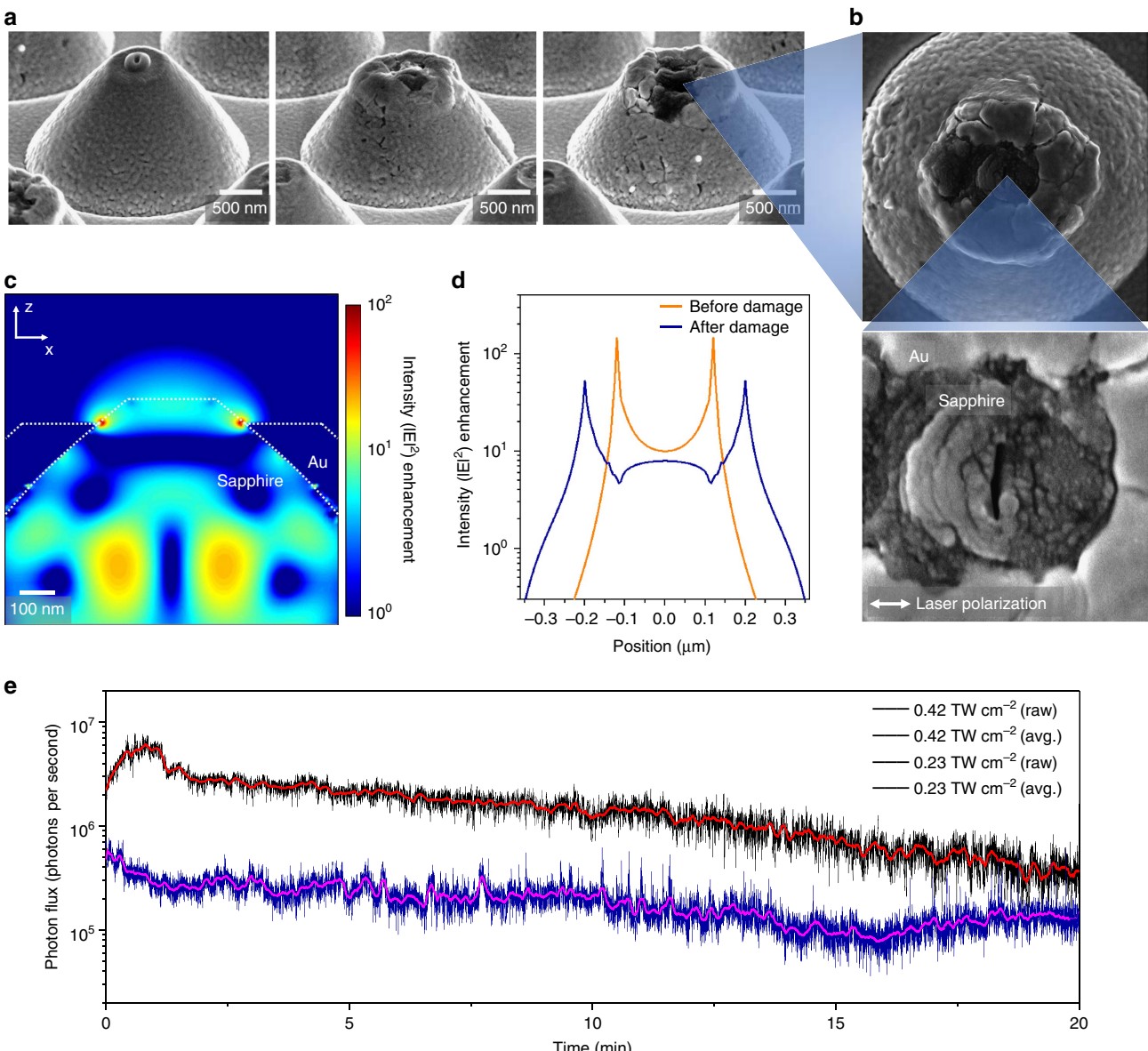

**Figure 3 | Thermal Damage.** (**a**) Scanning electron microscopy images of deformed sapphire tips for incident intensities of 0.23, 0.42 and 0.66 TW cm$^{-2}$, respectively. (**b**) Excessive thermal damage accompanied by surface cracks formed perpendicular to the direction of the incident field polarization. (**c**) FDTD calculation for the enhanced intensity distribution after thermal deformation of the tip shape. (**d**) Enhanced intensity profiles across the sapphire tip on the xz-plane before and after damage. (**e**) Variation of the extreme ultraviolet photon flux (H7, H9, H11 and H13) from the metal-sapphire nanostructure with increasing the laser exposure time.

gold-sapphire interface ends. The resulting intensity enhancement factor reaches ~20 dB. The electric field on the sapphire tip not only has the $x$ axis component but also a $z$ axis component (Fig. 1e), diffracting into a large conic angle as it leaves the exit aperture. The enhanced field calculated by FDTD simulation has almost the same temporal profile as the incident laser field without significant plasmonic dephasing (Fig. 2c).

**HHG experiment**. Figure 2a illustrates the spectrometer system used for the experiment in this study (see Methods), and Fig. 2b shows the extreme ultraviolet spectra observed from the metal-sapphire nanostructure. The incident laser field for experiment was produced from a Ti:sapphire oscillator emitting 12 fs pulses with an 800 nm center wavelength at a 75 MHz repetition rate. The laser field was focused on a ~5 µm spot diameter on the

bottom side of the nanostructure. The polarization direction of the incident laser field was set parallel to the sapphire C-plane and at the same time perpendicular to the sapphire A-plane. The laser average power was increased from 10 to 100 mW, while the resulting spectra were monitored over the wavelength range of 40–130 nm. The measured spectra (Fig. 2b) show that the seventh harmonic (H7) begins to appear when the incident laser power exceeds 40 mW. When the laser power reaches 75 mW, the appearance of harmonics extends to H13, for which the incident intensity is estimated to be 0.42 TW cm$^{-2}$ with 1.0 nJ energy per pulse. Most of the peaks in the measured spectra fall on the odd-order harmonics of the fundamental carrier wavelength of 800 nm. The bandgap of sapphire is known to be ~9 eV in the literature[29], which is not clearly observable in Fig. 2b since its emission line is located below the position of H7. The measured harmonics peaks of H7, H9 and H11 are well fitted by the

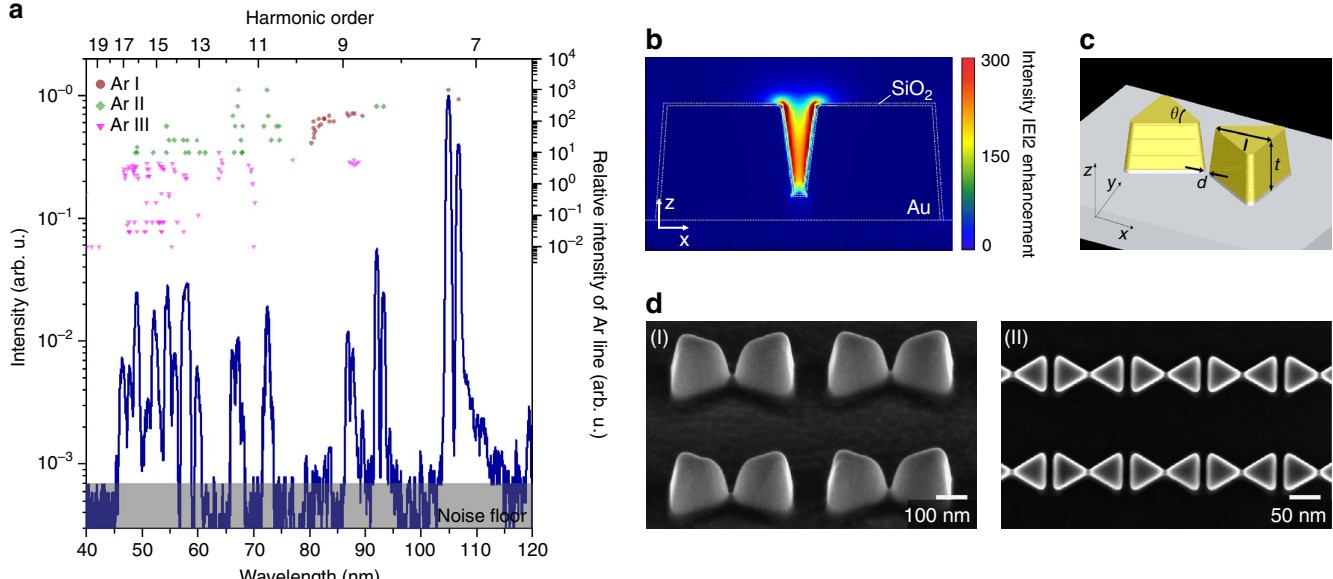

**Figure 4 | Extreme ultraviolet generation from bow-tie nanostructures with injection of Ar gas.** (**a**) Measured extreme ultraviolet spectrum. Plasma atomic lines of Ar are shown with their relative intensities. (**b**) FDTD calculation result of the enhanced intensity field in a bow-tie nanostructure. (**c**) Dimensional parameters; $t = 135$ nm, $l = 175$ nm, $\theta = 60°$ and $d = 15$ nm. (**d**) SEM images of bow-ties formed in a two-dimensional array; perspective (I) and top (II) views.

same Gaussian profile having a 0.06 eV normalized bandwidth (Fig. 2d). The normalized bandwidth denotes the photon energy spread $\triangle E$ of each harmonic peak divided by its harmonic order $N$, corresponding to the energy bandwidth of the incident laser field involved in the HHG. It is also noted that the profiles and locations of H7, H9 and H11 well agree with the harmonic peaks produced from a bare C-plane sapphire specimen using the same laser but with higher driving intensities (see Supplementary Fig. 1, Supplementary Note 1).

### Discussion

Figure 3 illustrates how the metal-sapphire nanostructure is affected by accumulated laser pulses. FDTD calculation estimates an intensity enhancement factor of $\sim 140$ on the edge side and $\sim 10$ on the central part of the sapphire tip with a 230 nm exit aperture (Fig. 1d). Such a strong field enhancement causes thermal damage even for weak input laser intensities of $\sim 0.1$ TW cm$^{-2}$. The initial sign of thermal damage is the surface cracking and/or deformation of the gold layer, while the dielectric sapphire remains solid (Fig. 3a). The consequence is that the exit aperture on the sapphire tip expands gradually to $\sim 400$ nm when the input laser intensity is set at 0.6 TW cm$^{-2}$. The FDTD calculation reveals that the intensity enhancement factor reduces to $\sim 53$ on the edge and $\sim 8$ in the central part (Fig. 3c), but the deformed shape is still robust enough to induce HHG with enhanced intensities of $\sim 4$ TW cm$^{-2}$. Further, the reduction in harmonic yield due to the decreased enhancement factor is compensated by the enlarged exit aperture acting as the HHG emitter, which produces harmonic peaks up to H13 as shown in Fig. 2b. Nonetheless, when the input intensity exceeds the threshold value of 0.6 TW cm$^{-2}$, the nanostructure no longer functions as an efficient HHG emitter since the gold layer appears to be severely damaged by melting and vapourizing. It is also noted that the sapphire tip begins to lose its initial shape due to the non-thermal photoemission of electrons and ions from the tip surface, which results in cracks that grow perpendicular to the laser polarization direction (Fig. 3b; ref. 30). In consequence, the

extreme ultraviolet yield from the metal-sapphire nanostructure varies with the exposure time as illustrated in Fig. 3e for two different driving laser intensities; 0.42 and 0.23 TW cm$^{-2}$. The photon flux for the higher intensity (0.42 TW cm$^{-2}$) reaches well above $10^6$ photons per second for a single nanostructure (see Methods) and gradually reduces because thermal damage develops with increasing the exposure time. On the other hand, the photon flux for the lower intensity (0.23 TW cm$^{-2}$) suffers less thermal damage and remains above $10^5$ photons per second throughout the total exposure time of 20 min. The conversion efficiency of a single nano-structure is calculated to be $10^{-11}$ for the dominant harmonic peak of H7, while it reduces by one order for H9 and H11, and roughly two orders for H13.

The generation mechanism of high-order harmonics from sapphire is not explained by the semi-classical three-step model[1] or the quantum model[2] commonly used for gaseous atoms. It is well known that the main distinct feature of HHG in crystalline solids such as sapphire is that the motion of electrons excited by tunnelling is confined to the band structure, and accordingly the dipole moment of electrons driven by the incident field is characterized by the inter- and/or intra-band oscillations[31–35]. For elaborate analysis of the HHG from solids, the band structure has to be modelled by two or multiple bands with energy levels varying with the polarization direction of the driving laser to the solid orientation[36]. The cutoff energy of the high-order harmonics generated from solids is known to linearly scale with the driving field strength[5,7], unlike in gaseous atoms where the cutoff energy is commonly in proportion with the intensity. This unique feature of solid-based HHG is also found in the case for the C-plane sapphire used as the emitter in our experiment (see Supplementary Fig. 2, Supplementary Note 2).

Our experimental result indicates that the HHG process from the sapphire tip begins to occur with enhanced intensities of $\sim 4$ TW cm$^{-2}$, which is lower than the threshold intensities of $\sim 10$ TW cm$^{-2}$ commonly reported for HHG from gaseous atoms[37]. The maximum enhanced intensity occurring on the edge

side of the exit aperture is estimated to reach $\sim 40\,\mathrm{TW\,cm^{-2}}$, and at this instance the electron excursion after tunnelling ionization is calculated to be $\sim 0.5$ nm (see Methods). The sapphire's lattice constant along the C-plane is 0.48 nm[38], which is comparable to the maximum electron excursion. This implies that the sapphire tip induces HHG dominantly by oscillations of electrons within a single crystalline structure. Another important fact is that the sapphire solid also acts as a strong extreme ultraviolet absorber, so only the surface layer of a several-lattice distance can function as an effective HHG emitter. The extreme ultraviolet radiation induced by HHG deep inside the nanostructure is re-absorbed immediately before escaping through the sapphire tip, resulting in no significant contribution to the harmonic yield. Therefore, the metal-sapphire nanostructure was devised to concentrate the field enhancement on the surface layer of the sapphire tip mainly within a 30 nm depth so that the extreme ultraviolet radiation can propagate through with a 0.05 transmission efficiency at least[39].

For comparison, Fig. 4 shows a spectrum obtained using gaseous atoms as the HHG emitter on a bow-tie nanostructure (see Methods). During experiment, Ar gas was injected on the bow-tie nanostructure through a 50 μm diameter nozzle with a 15 bar backing pressure. The measured extreme ultraviolet spectrum (Fig. 4a) appears to be strongly influenced by the plasma emission lines of the neutral atoms (Ar I) and singly- and doubly charged ions (Ar II and Ar III) as listed in the literature[40–42]. In contrast, HHG peaks are seen not sufficiently built up. The reason is reckoned that the number of gaseous atoms interacting with the enhanced field is not high enough, whereas the high repetition rate of the incident laser pulses at 75 MHz tends to provoke plasma formation resulting in strong fluorescent emission[27]. This implies that the HHG interpretation of ref. 18 made with the neglect of fluorescent emission was not fully correct as refuted later by similar bow-tie nanostructure experiments[25,26].

In conclusion, the metal-sapphire nanostructure devised in this study enabled us to demonstrate efficient HHG by field enhancement through a monocrystalline sapphire emitter. Unlike our previous approaches relying on gaseous atoms as the HHG emitter[18–20], the dielectric solid emitter permits coherent odd-order extreme ultraviolet harmonics to be generated without the dominance of fluorescence lines. In addition, the metal-sapphire nanostructure was found to be less susceptible to thermal damage than purely metallic nanostructures and its deformed structure maintains the field enhancement ability. These findings are believed to set an important milestone in the development of nanostructure-assisted HHG, which will inspire more experimental and theoretical investigations to realize the goal of robust, compact nano-scale extreme ultraviolet applications for microscopy, lithography, spectroscopy and even atto-second physics.

## Methods

**FDTD simulation.** The field enhancement on the sapphire tip was calculated using a commercial software program (Lumerical FDTD Solutions 8) by modelling the nanostructure, before and after damage, with cubic elements of $5.0 \times 5.0 \times 5.0$ nm dimensions. The electric permittivity values of sapphire, gold and vacuum were taken from the handbook[43]. The incident laser pulse was assumed to be transform-limited, having a 375 THz center frequency with 12 fs pulse duration. The time-domain computation was implemented with an incremental step of 0.0095 fs. Symmetrical boundary conditions about the $x$ and $y$ axis were adopted to reduce the computation time.

**Spectrometer system for experiment.** Extreme ultraviolet light is strongly absorbed by gas molecules in air, thus its generation with subsequent uses requires a vacuum environment of $10^{-4}$ Torr or less. Figure 2a shows the vacuum chamber system configured for our experiment, which comprises three sub-chambers; source chamber, intermediate chamber and detector chamber. The source chamber accommodates the nanostructure to be illuminated by the source laser. The intermediate chamber is inserted to install a toroidal mirror so that the extreme ultraviolet radiation emitted from the nanostructure is collected with an acceptance angle of 5° horizontal and 8° vertical, and subsequently refocused on to the entrance-slit aperture of the spectrometer grating. The detector chamber is equipped with a micro-channel plate attached to a charge-coupled detector to monitor the power spectrum of the extreme ultraviolet radiation along the Rowland circle of the spectrometer grating. The overall resolving power is estimated to be 0.1 nm in wavelength.

**Electron excursion.** When free electrons in an unoccupied conduction band are driven by a laser field, the maximum excursion is estimated as $R_{\mathrm{max}} = eE\lambda^2/4\pi^2mc^2$ in which $e$ is the electric charge of an electron, $E$ is the electric field of the incident laser, $\lambda$ is the wavelength of the laser, and $m$ is the effective mass of an electron[44]. For an input laser intensity of 0.42 TW cm$^{-2}$, or 1.8 V nm$^{-1}$ in terms of the electric field, the corresponding $R_{\mathrm{max}}$ works out to be 0.057 nm when $m$ is supposed to be the electron mass in vacuum. The metal-sapphire nanostructure produces a maximum field enhancement on the edge of the sapphire tip of a factor of $\sim 10$, which increases $R_{\mathrm{max}}$ to $\sim 0.5$ nm.

**Photon flux measurement.** The extreme ultraviolet yield was monitored using a Cu–BeO photomultiplier (R595, Hamamatsu) placed at a 20 mm distance from the nanostructure tip with an acceptance angle of 33° horizontal and 28° vertical. The photomultiplier responds to the wavelength range of 30–150 nm, which embraces high harmonics from the seventh to thirteenth order. By detecting the photocurrent, $I$, using a pico-ammeter (6,485, Keithley), the photon flux, $N$, is estimated by the formula of $N = I/(q \times Q \times G \times \eta)$, in which $q$ is the electron charge ($1.6 \times 10^{-19}$ C), $Q$ is the quantum efficiency of the photomultiplier (0.17), $G$ is the amplification gain of the pico-ammeter ($4 \times 10^5$) and $\eta$ is the collection efficiency (0.5) that is smaller than unity due to the light diffraction at the nanostructure tip.

**Bow-tie nanostructure design and experiment.** The bow-tie nanostructure used for the experiment described in Fig. 4 was designed to provide enhanced intensities of $\sim 10$ TW cm$^{-2}$ or stronger by means of resonant plasmonic field enhancement (Fig. 4b). The dimensions of a single bow-tie nanostructure are thickness $t = 135$ nm, length $l = 175$ nm, vertex angle $\theta = 6\hat{0}$ and gap distance $d = 15$ nm (Fig. 4c). During the experiment, the incident laser was focused on a target area comprising 500 bow-tie nanostructures arranged in a 2-D array with 500 and 600 nm spacing in the lateral $x$ and $y$ direction, respectively (Fig. 4d). The bow-tie structure was fabricated using the focused-ion-beam process on a sapphire substrate covered with an Au thin-film layer of 135 nm on a 3 nm Cr adhesive underneath layer. The fabricated specimen was coated with a SiO$_2$ thin film of 5 nm thickness to strengthen its thermal durability against melting. The main chamber was held in a $10^{-3}$ Torr vacuum condition with an injection of Ar gas, while the detector chamber was maintained at $10^{-6}$ Torr vacuum by differential pumping. The absolute wavelength position of the used spectrometer system was calibrated with reference to well-identified plasma lines of He, Ne and Ar gases.

**Data availability.** The data that support the finding of this study are available from the corresponding author upon request.

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

## Acknowledgements

This work was supported by the National Research Foundation of the Republic of Korea (NRF-2012R1A3A1050386). We recognize D.-H. Lee and J. Choi for their work setting up the spectrometer system and appreciate T. Lee for support in nanostructure fabrication at the KAIST Analysis Center for Research Advancement. Y.-J.K. acknowledges support from the Singapore National Research Foundation (NRF-NRFF2015-02). S.K. acknowledges support from the Ministry of Education of the Republic of Korea (NRF-2013R1A1A2004932).

## Author contributions

The project was planned and overseen by S.-W.K. in collaborations with Y.-J.K., S.K. and I.-Y.P. Nanostructure preparation and experiments were performed by S.H., H.K. and Y.W.K. All authors contributed to the manuscript preparation.

## Additional information

**Competing financial interests:** The authors declare no competing financial interests.

