## [Peer Review File · Nature Communications]

Reviewers' comments:

Reviewer #1 (Remarks to the Author):

Since 2008 many leading groups are trying to realize HHG by atoms close to nanotip structures or directly from nanotips. These efforts failed so far, mainly because nanostructures melt before generating sufficiently strongly enhanced plasmonic fields.

The present paper seems to overcome this problem. The authors propose plasmonic high harmonic generation (HHG) directly from a designed metal-sapphire nanostructure that provides a solid tip instead of gaseous atoms. "The fabricated solid tip is made of monocrystalline sapphire surrounded by a gold thin-film layer, and intended to produce coherent extreme ultraviolet (EUV) harmonics by the inter- and intra-band oscillations of electrons driven by the incident laser. Measured EUV spectra show odd-order harmonics up to ~60 nm wavelengths without the plasma atomic lines typically seen when using gaseous atoms as the HHG emitter. Nanostructure damage is under control

This result is truly amazing and I recommend publication in Nature Comm. This is a very important confirmation that plasmonic HHG is a promising way to realize coherent and efficient EUV sources.

As I said I recommend publication, but I suggest that the authors add /expand the discussion of the mechanism of the present process: is there anything common here or not with the "standard three step models" as described in the "classical HHG" papers by Corkum, Kulander or the quantum quasi-classical model Lewenstein et al.

Reviewer #2 (Remarks to the Author):

In the manuscript high-order harmonic generation (HHG) in metal-sapphire nanostructures illuminated with moderate intensity laser pulses from a Ti:sapphire oscillator is investigated. By employing a conical waveguide field intensities sufficient for harmonic generation could be achieved by means of surface plasmon polariton adiabatic nanofocusing. Unlike the previous work from the same group where extreme ultraviolet (EUV) emission was induced in atomic gas by enhanced nanoplasmonic fields in this work monocrystalline sapphire serves as the generating medium. Due to the much higher density of the solid coherent EUV generation is favored over incoherent processes such as atomic line emission. With this approach authors observed EUV emission peaked at odd-order harmonics of the driving laser field extending up to the 13-th harmonic order. The harmonic generation process was studied up to the intensities where significant deformation of the nanostructures is observed. This is an original work that presents an improved scheme for HHG in plasmonic nanostructured materials. It is an essential step forward from the previous work where atomic line emission dominated the observed EUV spectra. The experimental results are convincing, clearly presented and thoroughly discussed. The manuscript can be recommended for publication in Nature Communications after addressing the following points:

- 1) The authors do not provide any information on the conversion efficiency of the HHG process in the nanostructures. Even a rough estimate and comparison with the HHG in gases and solids would be interesting.
- 2) As is well known from the previous works on HHG in solids (ref 5, 6, 7) the cutoff energy of the high-order harmonics linearly scales with the strength of the driving field. This is distinctly different from harmonic generation in gas where the cutoff energy linearly scales with the intensity. Though it might be difficult to extract the exact scaling due to the nanostructure deformation and decrease of

the field enhancement at high laser intensities the cutoff scaling should be discussed in the paper.

3) Since noticeable deformation of the nanostructures was observed for most of the studied intensities it would be interesting to know how long the nanostructures can sustain the HHG process at these intensities without significant drop in the harmonic yield.

4) The authors should provide more details on how the experimental plot in Figure 2 (b) was obtained. In particular, information on the acquisition time, any smoothing or averaging should be included.

5) The authors might consider changing the color scale in Figure 1 (d) and Figure 3 (c) to avoid saturation.

Reviewer #3 (Remarks to the Author):

Nanojoule femtosecond laser pulses from a Ti:sapphire laser oscillator are focused onto sapphire tips covered with gold apertures to generate EUV radiation with wavelengths as short as 60 nm. The authors attribute this effect to plasmon-enhanced high-harmonic generation in the sapphire nanotips.

This topic is of potential interest to the readers, yet, I am afraid more evidence is needed to convince the readership of the microscopic picture. My scepticism regarding claims of the sort made in the present manuscript was created by some of the authors themselves: Reference 18 assigned fluorescence lines to high-harmonic radiation, which later turned out to be incorrect (see reference 25). In order to avoid similar controversy about the current manuscript, the authors should collect additional data. In particular, I would like to see how the frequencies of the harmonics change when the centre frequency of the driving field is modified. Also it would be good to show other harmonic orders as well.

Furthermore I am not convinced, yet, that the EUV radiation really originates from the sapphire alone. How can the authors exclude nonlinearities in the gold film (see e.g. Optical Materials Express 5, 2217 (2015))? The comparison of the present data with fluorescence spectra obtained with quite different bowtie antenna arrays does not seem to contain meaningful information in this respect.

On page 5 and in the Methods, the authors calculate the "electron excursion of free electrons to be ~ 0.5 nm" and, from this, they deduce that "the sapphire tip induces HHG dominantly by oscillations of electrons within a single crystalline structure." First it is not clear which "oscillations" the authors are referring to. Bloch oscillations? Second, the idea of transferring classical free electron motion to the solid state is naïve and not state of the art any more as shown in the theory references cited. Especially, there are prominent interband transitions on top of the semiclassical wavepacket motion as shown in Vampa et al., Nature 522, 462 or Hohenleuter et al., Nature 523, 572. A somewhat more rigorous description is required.

On a less fundamental note, it would be good to know what the conversion efficiency for 7th, 9th, 11th, and 13th harmonic generation is. What is the power in each harmonic order measured?

In the introduction as well as in the conclusions, the authors do not convey the fact that reference 18 claimed the wrong mechanism. I do not want to force the authors to explicitly state this fact here, but it is not okay to mislead the non-expert reader by citing this paper without additional comments. If the authors want to avoid a clear statement, I recommend removing reference 18 altogether from the reference list.

One final detail: There is a typo in the caption of figure 2 which says "Figure 1".

List of Changes: High harmonic generation by strongly enhanced femtosecond pulses in metal-sapphire nanostructure

We much appreciate the reviewers' comments and have done our best to address them all faithfully as summarized one by one below. (Original comments are reproduced in black, and our responses are written in blue.)

Reviewer #1 (Remarks to the Author):

Since 2008 many leading groups tried to realize HHG by atoms close to nanotip structures or directly from nanotips. These efforts failed so far, mainly because nanostructures melt before generating sufficiently strongly enhanced plasmonics fields.

The present paper seems to overcome this problem. The authors propose plasmonic high harmonic generation (HHG) directly from a designed metal-sapphire nanostructure that provides a solid tip instead of gaseous atoms. "The fabricated solid tip is made of monocrystalline sapphire surrounded by a gold thin-film layer, and intended to produce coherent extreme ultraviolet (EUV) harmonics by the inter- and intra-band oscillations of electrons driven by the incident laser. Measured EUV spectra show odd-order harmonics up to ~60 nm wavelengths without the plasma atomic lines typically seen when using gaseous atoms as the HHG emitter. Nanostructures damage is under control.

This result is truly amazing and recommend publication in Nature Comm. This is a very important confirmation that plasmonic HHG a promising way to realize coherent and efficient EUV sources. As i said I recommend publication, but i suggest that the authors add /expand the discussion of the mechanism of the present process: is there anything common here or not with the "standard three step models ad described the the "classical HHG" papers by Corkum, Kulander or the quantum quasi-classical model Lewenstein et al.

In response to the above reviewer's suggestion, the mechanism of the HHG process from sapphire has been discussed in the revised manuscript as

(page 5, line 25):

"The generation mechanism of high-order harmonics from sapphire is not explained by the semi-classical three-step model¹ or the quantum model² commonly used for gaseous atoms. It is well known that the main distinct feature of HHG in crystalline solids such as sapphire is that the motion of electrons excited by tunneling is confined to the band structure, and accordingly the dipole moment of electrons driven by the incident field is characterized by the inter- and/or intra-band oscillations³¹⁻³⁵. For elaborate analysis of the HHG from solids, the band structure has to be modelled by two or multiple bands with energy levels varying with the polarization direction of the driving laser to the solid orientation³⁶. The cutoff energy of the high-order harmonics generated from solids is known to linearly scale with the driving field strength^{5,7}, unlike in gaseous atoms where the cutoff energy is commonly in proportion with the intensity. This unique feature of solid-based HHG is also found in the case for the C-plane sapphire used as the emitter in our experiment (see Supplementary Information)."

(Supplementary Information):

"In this Supplementary Information, experimental HHG data obtained from a bare C-plane sapphire is presented and compared with the HHG spectra from the metal-sapphire nanostructure described in the main manuscript. Firstly, Supplementary Fig. 1(a) shows the HHG spectra monitored directly from a bare sapphire specimen that has no metallic nanostructure for field enhancement at all. The specimen was made of single crystal C-plane sapphire in the shape of a thin plate with a 430 μm thickness. The driving laser was focused directly on a 5 μm diameter spot over the specimen with increasing the average power to 280 mW. The polarization direction of the driving laser was aligned to be parallel to the C-plane and normal to the A-plane of the sapphire specimen. High harmonics up to the 13th order were clearly observed when the focused intensity reaches 1.47 TWcm^{-2} . Secondly, Supplementary Fig. 1(b) presents the cutoff energy (eV)

level of the HHG signals, which linearly scales with the input electric field as noted in the previous HHG works on solids. The cutoff energy was determined as the maximum harmonic peak discernible in each EUV spectrum of Supplementary Fig. 1(a).

Supplementary Figure 1 | HHG generation and energy cutoff estimation in bare C-plane sapphire. (a) Measured EUV spectra with increasing the field intensity. **(b)** Cutoff energy vs. electric field of the driving laser. The cutoff energy for each electric field is given by a blue dot with error bar. The red line is the linear least-square fitting of measured data.

(Reference 36; newly added)

Hohenleutner, M. *et al.* Real-time observation of interfering crystal electrons in high-harmonic generation. *Nature* **523**, 572–5 (2015).

Reviewer #2 (Remarks to the Author):

In the manuscript high-order harmonic generation (HHG) in metal-sapphire nanostructures illuminated with moderate intensity laser pulses from a Ti:sapphire oscillator is investigated. By employing a conical waveguide field intensities sufficient for harmonic generation could be achieved by means of surface plasmon polariton adiabatic nanofocusing. Unlike the previous work from the same group where extreme ultraviolet (EUV) emission was induced in atomic gas by enhanced nanoplasmonic fields in this work monocrystalline sapphire serves as generating medium. Due to much higher density of the solid coherent EUV generation is favored over incoherent processes such as atomic line emission. With this approach authors observed EUV emission peaked at odd-order harmonics of the driving laser field extending up to the 13-th harmonic order. The harmonic generation process was studied up to the intensities where significant deformation of the nanostructures is observed.

This is an original work that presents an improved scheme for HHG in plasmonic nanostructured materials. It is

an essential step forward from the previous work where atomic line emission dominated to the observed EUV spectra. The experimental results are convincing, clearly presented and thoroughly discussed. The manuscript can be recommended for publication in Nature Communications after addressing the following points:

1) The authors do not provide any information on the conversion efficiency of the HHG process in the nanostructures. Even rough estimate and comparison with the HHG in gases and solids would be interesting.

As requested, the conversion efficiency has been discussed as

(page 5, line 22)

“ ... The conversion efficiency of a single nanostructure is calculated to be 10^{-11} for the dominant harmonic peak of H7, while it reduces by one order for H9 and H11, and roughly two orders for H13.”

(page 8, line 17)

*“**Photon flux measurement.** The EUV yield was monitored using a Cu-BeO photomultiplier (R595, Hamamatsu) placed at a 20 mm distance from the nanostructure tip with an acceptance angle of 33° horizontal and 28° vertical. The photomultiplier responds to the wavelength range of 30 - 150 nm, which embraces high harmonics from the 7th to 13th order. By detecting the photocurrent, I , using a pico-ammeter (6485, Keithley), the photon flux, N , is estimated by the formula of $N = I / (q \times Q \times G \times \eta)$, in which q is the electron charge (1.6×10^{-19} C), Q is the quantum efficiency of the photomultiplier (0.17), G is the amplification gain of the pico-ammeter (4×10^5) and η is the collection efficiency (0.5) that is smaller than unity due to the light diffraction at the nanostructure tip.”*

2) As is well known from the previous works on HHG in solids (ref 5, 6, 7) the cutoff energy of the high-order harmonics linearly scales with the strength of the driving field. This is distinctly different from harmonic generation in gas where the cutoff energy linearly scales with the intensity. Though it might be difficult to extract the exact scaling due to the nanostructure deformation and decrease of the field enhancement at high laser intensities the cutoff scaling should be discussed in the paper.

The comment has been addressed by discussing the HHG data obtained from the bare C-plane sapphire as

(page 6, line 5)

“ ... The cutoff energy of the high-order harmonics generated from solids is known to linearly scale with the driving field strength^{5,7}, unlike in gaseous atoms where the cutoff energy is commonly in proportion with the intensity. This unique feature of solid-based HHG is also found in the case for the C-plane sapphire used as the emitter in our experiment (see Supplementary Information). “

(Supplementary Information):

“In this Supplementary Information, experimental HHG data obtained from a bare C-plane sapphire is presented and compared with the HHG spectra from the metal-sapphire nanostructure described in the main manuscript. Firstly, Supplementary Fig. 1(a) shows the HHG spectra monitored directly from a bare sapphire specimen that has no metallic nanostructure for field enhancement at all. The specimen was made of single crystal C-plane sapphire in the shape of a thin plate with a 430 μm thickness. The driving laser was focused directly on a 5 μm diameter spot over the specimen with increasing the average power to 280 mW. The polarization direction of the driving laser was aligned to be parallel to the C-plane and normal to the A-plane of the sapphire specimen. High harmonics up to the 13th order were clearly observed when the focused intensity reaches 1.47 TWcm^{-2} . Secondly, Supplementary Fig. 1(b) presents the cutoff energy (eV) level of the HHG signals, which linearly scales with the input electric field as noted in the previous HHG works on solids. The cutoff energy was determined as the maximum harmonic peak discernible in each EUV spectrum of Supplementary Fig. 1(a).

Supplementary Figure 1 /HHG generation and energy cutoff estimation in bare C-plane sapphire. (a) Measured EUV spectra with increasing the field intensity. (b) Cutoff energy vs. electric field of the driving laser. The cutoff energy for each electric field is given by a blue dot with error bar. The red line is the linear least-square fitting of measured data.

3) Since noticeable deformation of the nanostructures was observed for most of the studied intensities it would be interesting to know how long the nanostructures can sustain the HHG process at these intensities without significant drop in the harmonic yield.

The durability of the nanostructure has been discussed as

(page 5, line 17)

“ ... the EUV yield from the metal-sapphire nanostructure varies with the exposure time as illustrated in Figure 3e for two different driving laser intensities; 0.42 TWcm⁻² and 0.23 TWcm⁻². The photon flux for the higher intensity (0.42 TWcm⁻²) reaches well above 10⁶ photons/s for a single nanostructure (see Methods) and gradually reduces because thermal damage develops with increasing the exposure time. On the other hand, the photon flux for the lower intensity (0.23 TWcm⁻²) suffers less thermal damage and remains above 10⁵ photons/s throughout the total exposure time of 20 min.”

(Figure 3e; newly added)

Caption | Variation of the EUV photon flux (H7, H9, H11 and H13) from the metal-sapphire nanostructure with increasing the laser exposure time.

(page 8, line 17)

“Photon flux measurement. The EUV yield was monitored using a Cu-BeO photomultiplier (R595, Hamamatsu) placed at a 20 mm distance from the nanostructure tip with an acceptance angle of 33° horizontal and 28° vertical. The photomultiplier responds to the wavelength range of 30 - 150 nm, which embraces high harmonics from the 7th to 13th order. By detecting the photocurrent, I, using a pico-ammeter (6485, Keithley), the photon flux, N, is estimated by the formula of $N = I / (q \times Q \times G \times \eta)$, in which q is the electron charge (1.6×10^{-19} C), Q is the quantum efficiency of the photomultiplier (0.17), G is the amplification gain of the pico-ammeter (4×10^5) and η is the collection efficiency (0.5) that is smaller than unity due to the light diffraction at the nanostructure tip.”

4) The authors should provide more details on how the experimental plot in Figure 2(b) was obtained. In particular, information on the acquisition time, any smoothing or averaging should be included.

The sampling condition has been provided in the caption of Figure 2 as

(page 15, in caption)

“ ... Each spectrum was acquired during 30 s and its raw data was low-pass filtered in the wavelength domain with a 1.0 nm cutoff for smoothening. ... ”

5) The authors might consider changing the color scale in Figure 1 (d) and Figure 3 (c) to avoid saturation.

The two figures (Fig.1d & Fig.3c) have been redrawn with a logarithmic color scale as:

Reviewer #3 (Remarks to the Author):

Nanojoule femtosecond laser pulses from a Ti:sapphire laser oscillator are focused onto sapphire tips covered with gold apertures to generate EUV radiation with wavelengths as short as 60 nm. The authors attribute this effect to plasmon-enhanced high-harmonic generation in the sapphire nanotips.

This topic is of potential interest to the readers, yet, I am afraid more evidence is needed to convince the readership of the microscopic picture. My scepticism regarding claims of the sort made in the present manuscript was created by some of the authors themselves: Reference 18 assigned fluorescence lines to high-harmonic radiation, which later turned out to be incorrect (see reference 25). In order to avoid similar controversy about the current manuscript, the authors should collect additional data. In particular, I would like to see how the frequencies of the harmonics change when the centre frequency of the driving field is modified. Also it would be good to show other harmonic orders as well. Furthermore I am not convinced, yet, that the EUV radiation really originates from the sapphire alone. How can the authors exclude nonlinearities in the gold film (see e.g. Optical Materials Express 5, 2217 (2015))? The comparison of the present data with fluorescence spectra obtained with quite different bowtie antenna arrays does not seem to contain meaningful information in this respect.

It is not possible right now to conduct HHG experiment by changing the center frequency of the driving field because our current laser source is not able to provide the capacity. Instead, the reviewer's concerns have been addressed by describing the HHG data obtained from the bare sapphire with relevant discussions in Supplementary Information as

In this Supplementary Information, experimental HHG data obtained from a bare C-plane sapphire is presented and compared with the HHG spectra from the metal-sapphire nanostructure described in the main manuscript. Firstly, Supplementary Fig. 1(a) shows the HHG spectra monitored directly from a bare sapphire specimen that has no metallic nanostructure for field enhancement at all. The specimen was made of single crystal C-plane sapphire in the shape of a thin plate with a 430 μm thickness. The driving laser was focused directly on a 5 μm diameter spot over the specimen with increasing the average power to 280 mW. The polarization direction of the driving laser was aligned to be parallel to the C-plane and normal to the A-plane of the sapphire specimen. High harmonics up to the 13th order were clearly observed when the focused intensity reaches 1.47 TWcm⁻². Secondly, Supplementary Fig. 1(b) presents the cutoff energy (eV) level of the

HHG signals, which linearly scales with the input electric field as noted in the previous HHG works on solids. The cutoff energy was determined as the maximum harmonic peak discernible in each EUV spectrum of Supplementary Fig. 1(a).

Supplementary Figure 1 / HHG generation and energy cutoff estimation in bare C-plane sapphire. (a) Measured EUV spectra with increasing the field intensity. (b) Cutoff energy vs. electric field of the driving laser. The cutoff energy for each electric field is given by a blue dot with error bar. The red line is the linear least-square fitting of measured data.

Supplementary Figure 2 / Comparison of measured EUV spectra between a bare C-plane sapphire and the metal-sapphire nanostructure.

Lastly, Supplementary Fig. 2 shows a comparison of EUV spectra; one is from the metal-sapphire nanostructure with a driving intensity of 0.42 TWcm⁻² as described in the main manuscript, and the other is from the bare sapphire with a stronger intensity of 1.58 TWcm⁻² as reproduced from Supplementary Fig. 1.

Both the EUV spectra exhibit harmonics up to H13. When the bare sapphire was illuminated by the same lower intensity given to the nanostructure, only weak harmonics of H7 and H9 were observed as in Supplementary Fig. 1. This confirms the effect of strong field enhancement on HHG occurring in the nanostructure. The EUV yield from the bare sapphire was about two orders of magnitude higher, which was attributed by the large emitting area of a 5 μm diameter; the emitting area of the nanostructure is restricted to a 0.4 μm diameter due to its nanostructure sapphire tip size. Despite the difference in the EUV yield, the two spectra shown in Supplementary Fig. 2 clearly demonstrate a strong resemblance, evidencing the HHG peaks from the nanostructure are not significantly influenced by fluorescence emission from either sapphire or gold. The peak seen between H7 and H9 in the nanostructure's spectrum is reckoned as an even-order harmonic of H8 generated by the steep spatial variation of the enhanced field or due to unknown noise.

On page 5 and in the Methods, the authors calculate the "electron excursion of free electrons to be ~ 0.5 nm" and, from this, they deduce that "the sapphire tip induces HHG dominantly by oscillations of electrons within a single crystalline structure." First it is not clear which "oscillations" the authors are referring to Bloch oscillations? Second, the idea of transferring classical free electron motion to the solid state is naïve and not state of the art any more as shown in the theory references cited. Especially, there are prominent interband transitions on top of the semiclassical wavepacket motion as shown in Vampa et al., Nature 522, 462 or Hohenleuter et al., Nature 523, 572. A somewhat more rigorous description is required.

In response to the above reviewer's suggestion, the mechanism of the HHG process from sapphire has been discussed in the revised manuscript as

(page 5, line 25):

"The generation mechanism of high-order harmonics from sapphire is not explained by the semi-classical three-step model¹ or the quantum model² commonly used for gaseous atoms. It is well known that the main distinct feature of HHG in crystalline solids such as sapphire is that the motion of electrons excited by tunneling is confined to the band structure, and accordingly the dipole moment of electrons driven by the incident field is characterized by the inter- and/or intra-band oscillations³¹⁻³⁵. For elaborate analysis of the HHG from solids, the band structure has to be modelled by two or multiple bands with energy levels varying with the polarization direction of the driving laser to the solid orientation³⁶. The cutoff energy of the high-order harmonics generated from solids is known to linearly scale with the driving field strength^{5,7}, unlike in gaseous atoms where the cutoff energy is commonly in proportion with the intensity. This unique feature of solid-based HHG is also found in the case for the C-plane sapphire used as the emitter in our experiment (see Supplementary Information)."

(Reference 36; newly added)

Hohenleutner, M. *et al.* Real-time observation of interfering crystal electrons in high-harmonic generation. *Nature* **523**, 572–5 (2015).

On a less fundamental note, it would be good to know what the conversion efficiency for 7th, 9th, 11th, and 13th harmonic generation is. What is the power in each harmonic order measured?

The conversion efficiency of the nanostructure has been discussed as

(page 5, line 17)

"... the EUV yield from the metal-sapphire nanostructure varies with the exposure time as illustrated in Figure 3e for two different driving laser intensities; 0.42 TWcm^{-2} and 0.23 TWcm^{-2} . The photon flux for the higher intensity (0.42 TWcm^{-2}) reaches well above 10^6 photons/s for a single nanostructure (see Methods) and gradually reduces because thermal damage develops with increasing the exposure time. On the other hand, the photon flux for the lower intensity (0.23 TWcm^{-2}) suffers less thermal damage and remains above

10^5 photons/s throughout the total exposure time of 20 min.”

(Figure 3e; newly added)

Caption | Variation of the EUV photon flux (H7, H9, H11 and H13) from the metal-sapphire nanostructure with increasing the laser exposure time.

(page 8, line 17)

“Photon flux measurement. The EUV yield was monitored using a Cu-BeO photomultiplier (R595, Hamamatsu) placed at a 20 mm distance from the nanostructure tip with an acceptance angle of 33° horizontal and 28° vertical. The photomultiplier responds to the wavelength range of 30 - 150 nm, which embraces high harmonics from the 7th to 13th order. By detecting the photocurrent, I , using a pico-ammeter (6485, Keithley), the photon flux, N , is estimated by the formula of $N = I / (q \times Q \times G \times \eta)$, in which q is the electron charge (1.6×10^{-19} C), Q is the quantum efficiency of the photomultiplier (0.17), G is the amplification gain of the pico-ammeter (4×10^5) and η is the collection efficiency (0.5) that is smaller than unity due to the light diffraction at the nanostructure tip.”

In the introduction as well as in the conclusions, the authors do not convey the fact that reference 18 claimed the wrong mechanism. I do not want to force the authors to explicitly state this fact here, but it is not okay to mislead the non-expert reader by citing this paper without additional comments. If the authors want to avoid a clear statement, I recommend removing reference 18 altogether from the reference list.

As the reviewer requested, a clear statement has been made in the revised manuscript as

(page 7, line 4)

“ ... This implies that the HHG interpretation of Ref. 18 made with the neglect of fluorescent emission was not fully correct as refuted later by similar bow-tie nanostructure experiments^{25,26}. ”

One final detail: There is a typo in the caption of figure 2 which says "Figure 1".

The typo that was occurred during the automatic PDF conversion process of online manuscript submission has been corrected.

REVIEWERS' COMMENTS:

Reviewer #1 (Remarks to the Author):

i am satisfied with the revisions and recommend the paper for publication in Nature Comm.

Reviewer #2 (Remarks to the Author):

The authors addressed the points raised by the referee and can be recommended for publication in Nature Communications.

A minor comment:

The vertical axis in the SI plots in Figure 1 (a) and Figure 2 has a.u. units. Usually a.u. is an abbreviation for atomic units. The authors might consider using arb.u. instead.

Reviewer #3 (Remarks to the Author):

The authors have satisfactorily answered almost all of my questions as well as the questions by the other reviewers as far as I can judge.

Especially the new data on HHG in bulk sapphire driven with nanojoule pulses directly out of a low-power titanium:sapphire laser oscillator are quite surprising and constitute an interesting finding in themselves. If the authors actually manage to do HHG without field enhancement I have no doubts that the HHG in the nanostructure originates from the sapphire. Therefore, I support publication in Nature Communications after the authors have appropriately addressed the following remaining issue:

As seen in Supplementary Figure 2, similar HHG spectra are obtained with pump intensities of 0.42 TWcm⁻² and 1.58 TWcm⁻² with and without the near-field enhancement, respectively. This ratio corresponds to a near-field intensity enhancement of 3.8 whereas the electric field enhancement is only 1.9. These are the relevant quantities. It might be that the actual punctual field enhancement is calculated to be a bit stronger at the edges of the metal structure, but those hot spots seem not to be relevant for the HHG and they have not been measured experimentally. An electric field enhancement of 1.9 is enough, in my view, for a proof of principle experiment. Deriving a claim such as "strongly enhanced femtosecond pulses" as done in the title, however, is strongly exaggerated. I recommend the authors to tone down the manuscript in this respect and change their title accordingly.

List of Changes: High harmonic generation by field enhanced femtosecond pulses in metal-sapphire nanostructure

We appreciate the reviewers' comments and have addressed them as summarized below. (Reviewers' comments are reproduced in black, and our responses are stated in blue.)

Reviewer #1 (Remarks to the Author):

i am satisfied with the revisions and recommend the paper for publication in Nature Comm.

Reviewer #2 (Remarks to the Author):

The authors addressed the points raised by the referee and can be recommended for publication in Nature Communications.

A minor comment:

The vertical axis in the SI plots in Figure 1 (a) and Figure 2 has a.u. units. Usually a.u. is an abbreviation for atomic units. The authors might consider using arb.u. instead.

As suggested, the abbreviation 'a.u.' used in the figures has been changed to 'arb. u.'.

Reviewer #3 (Remarks to the Author):

The authors have satisfactorily answered almost all of my questions as well as the questions by the other reviewers as far as I can judge.

Especially the new data on HHG in bulk sapphire driven with nanojoule pulses directly out of a low-power titanium:sapphire laser oscillator are quite surprising and constitute an interesting finding in themselves. If the authors actually manage to do HHG without field enhancement I have no doubts that the HHG in the nanostructure originates from the sapphire. Therefore, I support publication in Nature Communications after the authors have appropriately addressed the following remaining issue:

As seen in Supplementary Figure 2, similar HHG spectra are obtained with pump intensities of 0.42 TWcm⁻² and 1.58 TWcm⁻² with and without the near-field enhancement, respectively. This ratio corresponds to a near-field intensity enhancement of 3.8 whereas the electric field enhancement is only 1.9. These are the relevant quantities. It might be that the actual punctual field enhancement is calculated to be a bit stronger at the edges of the metal structure, but those hot spots seem not to be relevant for the HHG and they have not been measured experimentally. An electric field enhancement of 1.9 is enough, in my view, for a proof of principle experiment. Deriving a claim such as "strongly enhanced femtosecond pulses" as done in the title, however, is strongly exaggerated. I recommend the authors to tone down the manuscript in this respect and change their title accordingly.

As recommended, a discussion has been added in the Supplementary Note 2 as

" ... Direct comparison of the driving intensities of the two spectra indicates that the nanostructure provides an average intensity enhancement factor of 3.8. It is reckoned that the strong field enhancement

at the edges of the nanostructure tip offers limited contributions to the measured spectrum because it is narrowly localized and resulting EUV radiation is subject to severe diffractive scattering beyond the acceptance angle of the spectrometer system used in our experiment.”

Accordingly, the title has been changed to *‘High harmonic generation by field enhanced femtosecond pulses in metal-sapphire nanostructure.’*

In addition, the word ‘strong’ in the conclusion has also been removed as

“In conclusion, the metal-sapphire nanostructure devised in this study enabled us to demonstrate efficient HHG by field enhancement through a monocrystalline sapphire emitter....”